# Nanomaterials in Electrochemical Sensing Area: Applications and Challenges in Food Analysis

**DOI:** 10.3390/molecules25235759

**Published:** 2020-12-07

**Authors:** Antonella Curulli

**Affiliations:** Istituto per lo Studio dei Materiali Nanostrutturati (ISMN) CNR, Via del Castro Laurenziano 7, 00161 Roma, Italy; antonella.curulli@cnr.it; Tel.: +39-06-4976-7643

**Keywords:** nanomaterials, electrochemical sensors, hydroxycinnamic acids, caffeine, nitrite

## Abstract

Recently, nanomaterials have received increasing attention due to their unique physical and chemical properties, which make them of considerable interest for applications in many fields, such as biotechnology, optics, electronics, and catalysis. The development of nanomaterials has proven fundamental for the development of smart electrochemical sensors to be used in different application fields such, as biomedical, environmental, and food analysis. In fact, they showed high performances in terms of sensitivity and selectivity. In this report, we present a survey of the application of different nanomaterials and nanocomposites with tailored morphological properties as sensing platforms for food analysis. Particular attention has been devoted to the sensors developed with nanomaterials such as carbon-based nanomaterials, metallic nanomaterials, and related nanocomposites. Finally, several examples of sensors for the detection of some analytes present in food and beverages, such as some hydroxycinnamic acids (caffeic acid, chlorogenic acid, and rosmarinic acid), caffeine (CAF), ascorbic acid (AA), and nitrite are reported and evidenced.

## 1. Introduction

The introduction of novel functional nanomaterials and analytical technologies indicate the possibility for advanced electrochemical (bio)sensor platforms/devices for a wide number of applications, including biological, biotechnological, clinical and medical diagnostics, environmental and health monitoring, and food industries.

Nanoscale materials and nanomaterials are known as materials where any measurement is not as much as 100 nm. Nanomaterials reveal exciting properties that make them appeal to be exploited in electrochemistry and in the improvement of the (bio)sensors. Recent advances in nanotechnology have created a growing demand for their possible commercial application [1].

Nanotechnology involves the synthesis and characterization of nanomaterials, whereby nanomaterial can be defined as a natural or synthesized material containing particles, in an unbound state or as an aggregate or as an agglomerate, and where, for 50% or more of the particles in the number size distribution, one or more external dimensions is in the size range of 1–100 nm [2,3].

By reducing the material dimensions at the nanometre level, the chemical and physical properties of such a material can be modified and they are totally different with respect to the same corresponding bulk material [4,5,6].

Carbon nanotubes (CNTs) and gold nanoparticles (AuNPs) are among the most broadly explored nanomaterials because of their exceptional properties, which can be connected in different applications, e.g., detecting, and imaging. Yet, to date, the exploration field of advancement for the synthesis of new functionalized AuNPs and CNTs for sensing applications is a dynamic research territory. The combination of these nanomaterials has been developed, promoting improvements in controlling their size and shape [7,8].

Electroanalytical methods and electrochemical sensors have improved the analytical approach in different application fields, ranging from the biomedical to the environmental ones [9].

Particularly the modification and/or functionalization of the electrodic surface with nanomaterials involves an amplification of the corresponding electrochemical signal and it has proven very attractive for developing sensors with high sensitivity and selectivity [9].

In this review, we present a survey of the applications of different nanomaterials and nanocomposites as electrochemical sensing platforms for food analysis. The sensors analytical parameters, such as linearity range, detection limit, selectivity, and their possible applications to real samples are described and highlighted.

## 2. Electrochemical Techniques

Electrochemistry offers a wide range of electroanalytical techniques. A typical electrochemical experiment includes a working electrode made of a solid conductive material, such as platinum, gold, or carbon, a reference electrode, and a counter electrode, all the electrodes are generally immersed in a solution with a supporting electrolyte to guarantee the conductivity in the solution [10].

Electrochemical sensors belong to the largest family of chemical sensors. A chemical sensor can be defined as ‘‘a small device that, as the result of a chemical interaction or process between the analyte and the sensor device, transforms chemical or biochemical information of a quantitative or qualitative type into an analytically useful signal” [11]. This definition can be extended to the electrochemical ones modifying it in this way: a small device that, as the result of an electrochemical interaction or process between the analyte and the sensing device, transforms electrochemical information of a quantitative or qualitative type into an analytically useful signal [11]. The use of a nanomaterial together with the analyte kind and nature have proven crucial for the sensor sensitivity, selectivity, and stability [12]. As for all the chemical sensors, the critical parameters of electrochemical sensors are sensitivity, detection limit, dynamic range, selectivity, linearity, response time, and stability [13].

Several electrochemical methods have been employed for the detection of food additives, biological contaminants, and heavy metals [9].

In general, an electrochemical reaction can generate different measurable data, depending on the electrochemical technique adopted. In fact, a measurable current can be generated, and in this case, the corresponding electrochemical techniques are the amperometric ones. Alternatively, a potential can be measured and/or controlled, and in this case, the corresponding electrochemical techniques are the potentiometric ones. Finally, the electrochemical techniques, involving measurements of impedance at the electrode/solution interface are included in the electrochemical impedance spectroscopy (EIS) method [14].

Starting from the presentation of EIS, we propose a brief presentation and overview of the best known and used electrochemical techniques.

EIS is an electroanalytical method used for the evaluation of electron-transfer properties of the modified surfaces and in understanding of surface chemical transformations. EIS analysis provides mechanistic and kinetic information on a wide range of materials, such as batteries, fuel cells, corrosion inhibitors, etc. [14].

The overall electrochemical behaviour of an electrode can be represented by an equivalent circuit comprising resistance, inductance, and capacitance. The equivalent circuit elements, useful for analyte detection are resistance to the charge transfer R_ct_ and the double layer capacitance C_dl_. The measured capacitance usually arises from the series combination of several elements, such as analyte binding (C_anal_) to a sensing layer (C_sens_) on the electrode (C_el_). The sensitivity is then determined by the relative capacitance of the analyte layer and the sensing layer. One difficulty with capacitive sensors is that their sensitivity depends on obtaining the proper thickness of the original sensing layer.

Voltammetry belongs to the class of the amperometric techniques because the current produced from an electrochemical reaction is measured whilst varying the potential window. Since there are many ways to vary the potential, we can consider many voltammetric techniques. Among others, the most common and employed are the following: cyclic voltammetry (CV), linear sweep voltammetry (LSV), differential pulse voltammetry (DPV), and square wave voltammetry (SWV) [15,16,17,18].

CV and LSV are widely employed voltammetric techniques to study the electrochemical behaviour of an electroactive molecule.

DPV and SWV can be classified as pulse voltammetric techniques.

DPV and SWV in comparison with CV can be used to study the redox properties of extremely small amounts of electroactive compounds for several reasons, but principally: (1) in these measurements, the effect of the charging current can be minimized, so higher sensitivity is achieved and (2) only faradaic current is extracted, so electrode reactions can be analyzed more precisely.

Chronoamperometry (CA) is a potentiostatic technique, where the current is recorded as a time function. In Figure 1, an overview of the electrochemical methods of analysis, namely voltammetry, amperometry, electrochemical impedance spectroscopy (EIS), and potentiometry, is reported.

All the above-mentioned techniques have been widely employed in the development of electrochemical sensors for different application fields.

## 3. Nanomaterials, Nanotubes, Nanoparticles, and Nanocomposites

Recent developments in nanomaterial synthesis have allowed the development of advanced sensing systems [9,19].

Generally, we have considered modifications of the working electrode with different nanomaterials, ranging from the classical nanotubes to nanocomposites, among different nanostructures, such as graphene, metal nanoparticles, and/or nanostructured polymers. In addition, non-conventional sensing platforms, such as paper and/or screen-printed electrodes (SPE), also modified with different nanomaterials and/or nanostructures, have been considered. Hence, in this review, we report some example of these non-conventional sensing platforms, present in the literature and employed for different application fields [20,21].

Nanomaterials play a crucial role in the development of electrochemical sensors, improving the sensor stability, sensitivity, and selectivity in the presence of the common interferences.

In the following subparagraphs we introduce and describe nanomaterials and examples of the related electrochemical sensors just to show their applicability in the electrochemical sensing area.

### 3.1. Carbon-Based Nanomaterials

Carbon-based nanomaterials (single-walled carbon nanotubes (SWNTs), multi-walled carbon nanotubes (MWNTs), single-walled carbon nanohorns (SWCNHs), buckypaper, graphene, fullerenes (e.g., C_60_), etc. present very interesting properties, such as high surface-to-volume ratio, high electrical conductivity, chemical stability/durability, and strong mechanical strength, and for these reasons they have found a large applicability in the sensing area [22,23,24,25,26,27,28].

Carbon nanotubes (CNTs) present several properties associated to their structure, functionality, morphology, and flexibility to be employed in synthesis of hybrid or composite materials due to their hollow cylindrical structure.

Carbon nanotubes can be classified as single-walled nanotubes (SWNTs), double-walled nanotubes (DWNTs), and multi-walled nanotubes (MWNTs) depending on the number of graphite layers. Functionalized CNTs have been used in several application fields. The chemical functionalities can easily be designed and tuned through the tubular structure modification.

Some interesting examples of carbon nanomaterials based electrochemical sensors, related to different application fields not only to food analysis, are illustrated below.

Venton and co-workers have used metal microelectrodes modified with CNTs for assembling an electrochemical sensor for detecting dopamine in vitro and in vivo. [29]. It has been found that CNTs-coated niobium (CNTs-Nb) microelectrode showed a low detection limit of 11 nM for dopamine. The CNTs-Nb sensor was also employed to detect stimulated dopamine release in anesthetized rats and showed high sensitivity for in vivo measurements.

The design and synthesis of functionalized CNTs for biological and biomedical applications are highly attractive because in vivo sensing requires high selectivity, accuracy, and long-term stability. Zhang et al. have prepared an electrochemical ascorbic acid sensor for measuring ascorbic acid in brain using aligned carbon nanotube fibers (CNF) as a microsensor [30], obtaining very interesting results. The sensor measured ascorbic acid concentration of 259.0 μM in the cortex, 264.0 μM in the striatum, and 261.0 μM in the hippocampus, respectively, under normal conditions.

Graphene is one of most applied nanomaterial in the sensing area. Different graphene-based materials have been produced (e.g., electrochemically and chemically modified graphene) using many procedures [31]. Graphene shows properties such as high conductivity, accelerating electron transfer, and a large surface area, very similar indeed to the corresponding properties of CNTS, so it is considered a good candidate for assembling sensors to determine several target molecules [9,31].

Graphene oxide (GO) is hydrophilic and can be dispersed in water solution because of hydrophilic functional groups (OH, COOH and epoxides) at the edge of the sheet and on the basal plane.

On the other hand, GO has a low conductivity in comparison to graphene, so reduced GO (rGO) is more employed as electrode modifier in electrochemical sensing/biosensing area [31].

Fluorine doped graphene oxide was used to prepare an electrochemical sensor for the detection of heavy metal ions such as Cd^2+^, Pb^2+^, Cu^2+^, and Hg^2+^. Square wave anodic stripping voltammetry was employed for the detection of the heavy metal ions. The authors have evidenced that the presence of fluorine builds a more appropriate platform for the stripping process, from the comparison between the sensor based on GO and the sensor based on F-GO [9].

Li and co-workers developed a based electrochemical sensor for the detection of metal ions, Pb^2+^ and Cd^2+^, employing a Nafion–graphene composite film. The synergistic effect of graphene nanosheets and Nafion gave rise to a better sensitivity for detecting metal ion and enhanced the electrochemical sensor selectivity [32].

Li and co-workers have reported a graphene potassium doped modified glassy carbon electrode, for the determination of sulphite in water solution. A linear response in the concentration range of 2.5 μM–10.3 mM with a detection limit of 1.0 μM for SO_3_^2−^ has been obtained. The graphene electronic properties resulted modified by the K doping [33].

A glassy carbon electrode was modified by means of hexadecyl trimethyl ammonium bromide (CTAB) functionalized GO/multiwalled carbon nanotubes (MWCNTs) for the detection of ascorbic acid (AA) and nitrite. The combination of GO and MWCNTs provided a sensing platform with high electrocatalytic activity, increased surface area, as well as good stability and sensitivity, allowing for the determination of nitrite and AA at the same time [34].

### 3.2. Gold Based Nanomaterials

Since the first examples of gold nanoparticles (AuNPs) synthesis, AuNPs have been employed for the assembly of different sensors. From an electroanalytical point of view, Au nanoparticles and in general gold nanomaterials were employed in the electrochemical sensing area because of their high conductivity, their compatibility, and a high surface to volume ratio [35,36]. Gold nanomaterials have been used for the selective oxidation processes, or rarely, for the reduction ones.

Important improvements have been performed in the Au nanoparticles and nanomaterials synthesis for electrochemical sensing. However, researchers are dealt with several challenges/criticalities, such as size control, morphology, and suitable dispersion and/or stabilizing agents and/or media. The recent development of nanoporous Au materials seems to be fundamental in overcoming these challenges and hierarchical gold nanoporous materials have been recently reported for biomedical applications [37]. Some interesting examples of gold nanomaterials based electrochemical sensors, related to different application fields not only to food analysis, are illustrated below.

An electrochemical sensor using a gold microwire electrode for the detection of heavy metals ions, such as copper and mercury, in seawater was described by Salaün and coworkers. The sensor using gold microwires was able to detect the two metals ions at the same time by means of anodic stripping voltammetry [38].

Concerning the same analytical issue, i.e., the detection of different heavy metals ions at the same time, Soares and co-workers developed an electrochemical sensor using a vibrating gold microwire, and determined arsenic, copper, mercury, and lead ions in freshwater by means of stripping voltammetry [39].

Recently, gold nanopores were synthesized by the alloying/dealloying method for increasing the electrochemical performance of an analyte under investigation and several examples are reported below.

Ding and co-workers prepared a nanoporous gold leaf by dealloying the Au/Ag alloy in nitric acid. The resulting 3D gold nanostructure allowed the small molecules movement inside. For example, it was employed to modify a GCE for detecting nitrite. The increased surface area and the high conductivity of the 3D nanostructured gold network justify the good performance of the sensor [40].

Other examples concerning the use of nanoporous gold have involved more properly the electrochemical investigation of different analytes with this new nanomaterial as proof of concept to apply it in the sensing area.

A green synthesis of nanoporous gold materials was proposed by Jia and co-workers by means of a cyclic alloying/de-alloying procedure. The resulting nanoporous gold film modified electrodes showed very interesting electrochemical performances in terms of a high surface area and good selectivity [41].

Lin et al. modified a GCE via the electrodeposition of Au nanoparticles on polypyrrole (PPy) nanowires. AuNPs enhanced the conductivity of the polymer nanowires and consequently the electron transfer rate resulted higher than that at bare GCE and/or at GCE just modified with the polymer nanowires [42].

Finally, a nanoporous Au 3D nanostructure was synthesized as a proof of concept to be applied as a sensing platform for detecting hydrazine, sulphite, and nitrite, present in the same sample. The nanostructured sensor showed good performances in terms of selectivity and sensitivity [43].

### 3.3. Hybrid Nanocomposites

To improve and amplify the performances of a sensor and/or a sensing platform, nanomaterials such as carbon and/or metal nanomaterials were incorporated in different polymers both natural (e.g., chitosan) or (electro)synthesized (e.g., PEDOT, polypyrrole). Some interesting examples of hybrid nanocomposite based electrochemical sensors, related to different application fields not only to food analysis, are illustrated below [44].

As a first example, we can introduce an electrochemical sensor using a polypyrrole–chitosan–titanium dioxide (PPy–CS–TiO_2_) nanocomposite for glucose detection. Interactions between the TiO_2_ nanoparticles and PPY enhanced the sensor properties in terms of sensitivity and selectivity [45].

Bimetallic Au–Pt nanoparticles have been incorporated in rGO and the electrochemical behaviour of resulting nanocomposite was investigated and compared with the electrochemical behaviour of the bimetallic nanoparticles and of the rGO. An enhanced electrochemical activity is observed, probably due to the increase of the surface area and to the increase of the nanocomposite conductivity respect to the bimetallic nanoparticles and to the rGO [46].

Feng et al. assembled a sensor for caffeic acid detection through a nanocomposite obtained by the combination of worm-like Au–Pd nanostructures and rGO [47]. From the spectroscopic characterization, the nanotubular worm-like Au–Pd nanostructures were uniformly distributed on rGO.

A carbon paste electrode was modified with a nanocomposite obtained by combining AuNPs and MWCNTs. The oxidation of nitrite was investigated at such a modified electrode and showed better performances in terms of electrocatalytic behaviour respect to those at the bare carbon paste electrode and/or to those at CPE modified with only AuNPs or with only MWCNTs [48].

A hybrid nanocomposite was prepared by assembling Pt nanoparticles on a graphene surface. The modified electrode was used for the detection of ascorbic acid (AA), uric acid (UA), and dopamine (DA), obtaining interesting results in terms of selectivity [49].

Chen and co-workers proposed a Pt nanocomposite combining Pt nanoparticles and single-walled carbon nanotubes (SWCNTs), instead of graphene or MWCNTs.

A GCE modified with this nanocomposite was used for the electrochemical detection of α-methylglyoxal. A good linearity in the concentration range of 0.1–100.0 μM, and a detection limit of 2.80 nM were obtained. The sensor was applied to detect α-methylglyoxal in real samples of wine and beer [50].

Yegnaraman and co-workers reported an Au based nanocomposite for the detection of AA, UA, and DA to test the selectivity for detecting analytes present in the same solution. The nanocomposite film was synthesized by introducing Au nanoparticles into the PEDOT polymer matrix. The modified GCE determined AA, UA, and DA simultaneously, with improved sensitivity and selectivity [51].

A glucose impedimetric biosensor [52] was assembled using a metal composite composed by a gold microtubes (AuμTs) architecture and polypyrrole overoxidized by Curulli and co-workers. A platinum (Pt) electrode was coated by gold microtubes, synthesized via electroless deposition within the pores of polycarbonate particle track-etched membranes (PTM). This platform was successfully used to deposit polypyrrole overoxidized film (OPPy) and to verify the possibility of developing a biosensor using OPPy, the characteristics of the H_2_O_2_ charge transfer reaction were studied before the enzyme immobilization. This composite material seems to be suitable in devices as biosensors based on oxidase enzymes, just because hydrogen peroxide is a side-product of the catalysis and could be directly related to the concentration of the analyte. Finally, a biosensor consisting of a Pt electrode modified with AuμTs, OPPy, and glucose oxidase was assembled to determine the glucose. The most important result of this biosensor was the wide linear range of concentration, ranging from 1.0 to 100 mM (18–1800 mg·dL^−1^), covering the hypo- and hyperglycaemia range, useful in diabetes diagnosis, with limit of detection of 0.1 mM (1.8 mg·dL^−1^) and limit of quantification of 1.0 mM (18 mg·dL^−1^).

A wide range of glucose biosensor prototypes have been developed during these years, but the challenge is to obtain a biosensor capable of measuring the glucose concentration in the normal range (i.e., representative of healthy patients) and in the range of hyperglycaemia values (especially for diabetes patients, for clinical diagnosis of this widespread pathology). Very few examples propose biosensors with improved analytical performances, especially in terms of an extended linearity (still 50 mM ≈ 901 mg·dL^−1^, useful for the hyperglycaemia pathology values), high selectivity toward the most common interferents and an improved stability [53,54,55].

## 4. Electrochemical Sensors for Food Analysis: Some Examples

In this section, several examples of the electrochemical sensors employing different nanomaterials and applied to detect different analytical targets such as some hydroxycinnamic acids (caffeic acid, chlorogenic acid, and rosmarinic acid), ascorbic acid, and nitrite were illustrated and discussed.

### 4.1. Phenolic Antioxidants

Natural antioxidants are species of great interest in many areas, as food chemistry, health care and clinical applications. They have beneficial effects on human health and could play an important role in the prevention and treatment of many pathologies (such as cardiovascular disorders, cancer, etc.) and to protect from oxidative stress [56,57,58,59,60,61]. The classification of antioxidants is commonly carried out based on the chemical structure, determining their reactivity. However, their antioxidant action is also strictly related to the redox properties and consequently their knowledge is very crucial for a better understanding of antioxidant mechanisms.

Among the natural antioxidants found in fruits and plants, hydroxycinnamic acids (HAs) are very important and present in all parts of the fruit and/or plant [62,63,64]. Undoubtedly, these compounds in food provide added value for their well-known health benefits, for their technological role, and marketing. The electrochemical methods have been extensively used to investigate the redox properties of various species and as analytical tool for the determination of redox target molecules. At present, as for other classes of antioxidants, the analysis of HAs and phenolic antioxidants is usually carried out using chromatographic techniques, which require sophisticated equipment and laborious analytical procedures [57]. The use of electrochemical methods for analytical purposes is receiving increasing interest [57], since they are fast, accurate, sensitive and can be used for the analysis of different and complex matrices with a low cost.

Both electrochemical sensors and biosensors are widely used for the determination of HAs and phenolic antioxidants. However, the electrochemical responses have been studied only from an analytical point of view, whereas the relationship between the antioxidant chemical structure and electrochemical behaviour has been neglected [65]. The understanding of key factors that affect the electrochemical response of analytes could promote the design of highly efficient sensors. To this aim, the role of the chemical structural features of antioxidants and nature of the electrode surface must be evaluated.

Recently, electrodes modified with nanocomposite films were successfully used for the analysis of antioxidants, such as caffeic acid, in complex matrices [66,67]. These films consist of gold nanoparticles (AuNPs) embedded into chitosan, a biodegradable and biocompatible polymer containing many hydroxyl and amino groups that can interact with the analytes. Later, the electrochemical response of structurally related antioxidants on electrodes modified with different gold–chitosan nanocomposite films was investigated and discussed [65,66].

To evaluate how the chemical structural features of analytes affect the electrochemical behaviour, different types of antioxidants, such as catechols, hydroxycinnamic acids, and flavonoids, structurally correlated and bearing different functional groups and steric hindrances, have been studied. This investigation [66] has demonstrated that the electrochemical response for structurally related antioxidants at AuNPs–chitosan modified electrodes depends on several parameters. The chemical structural features of the analytes affect the interaction with the electrode surface. However, their electrochemical behaviour cannot be explained only on these bases.

The nanostructure and surface functional groups of AuNPs-chitosan modified electrodes have also a key role. In particular, the formation of a collaborative network with interconnected metal nanoparticles in chitosan film significantly affects the electron transfer properties, whereas the surface functional groups can promote the interaction with the antioxidants. An overview of the behaviour of catechols, hydroxy cinnamic acids, and flavonoids derivatives at different AuNPs–chitosan modified electrodes has been illustrated by Curulli and co-workers [66].

A better response was observed for molecules with two hydroxyl groups in ortho position of the catechol ring, with a peculiar molecular symmetry (i.e., rosmarinic acid) and with a low steric hindrance, in particular for caffeic acid, chlorogenic acid, and rosmarinic acid. Moreover, the interaction with the antioxidants is also affected by the functional groups at modified electrode surface and the size and distribution of AuNPs into the polymeric matrix. The understanding of the parameters affecting the electrochemical behaviour of analytes at modified electrodes is a key issue and could significantly promote the design of highly efficient sensors.

#### 4.1.1. Caffeic Acid

Caffeic acid (CA, 3,4-dihydroxycinnamic acid, chemical structure in Figure 2) is well known as a phenolic antioxidant. CA is present in wines, coffee, olive oil, as well as in some vegetables and fruits. It has several pharmacological functions and properties [62,68].

According to the literature [66], several analytical methods have been employed for the detection of caffeic acid, from liquid chromatography to the electrochemical ones. Considering the electrochemical approach, involving nanomaterials, several examples are reported in Table 1.

Gold–chitosan nanocomposite is successfully proposed for assembling a sensitive and selective electrochemical sensor for the determination of an antioxidant such as caffeic acid by Curulli’s group [66,67]. Taking advantage of the peculiar sensing performance of the nanocomposite, an analytical method based on differential pulse voltammetry for the determination of the polyphenol index in wines was proposed.

In this approach, colloidal gold nanoparticles (AuNPs) stabilized into a chitosan matrix were prepared using a green route. The synthesis was carried out by reducing AuIII to Au0 in an aqueous solution of chitosan and different organic acids. It has been demonstrated that by varying the nature of the acid it is possible to tune the reduction rate of the gold precursor (HAuCl_4_) and to modify the morphology of the resulting metal nanoparticles. The use of chitosan enables the simultaneous synthesis and surface modification of AuNPs in one pot. Because of the excellent film-forming capability of this polymer, AuNPs−chitosan solutions were used to obtain hybrid nanocomposite films that combine highly conductive AuNPs with many organic functional groups. Figure 2, the proposed scheme for the interaction between the AuNPs-chitosan nanocomposite and caffeic acid is illustrated.

Au-chitosan nanocomposites are successfully proposed as sensitive and selective electrochemical sensors for the determination of caffeic acid, by means of Differential Pulse Voltammetry. A linear response was obtained over a wide range of concentration from 5.00 × 10^−8^ M to 2.00 × 10^−3^ M, and the limit of detection was 2.50 × 10^−8^ M. Moreover, further analyses have demonstrated that a high selectivity toward caffeic acid can be achieved without interference from catechin or ascorbic acid (flavonoid and nonphenolic antioxidants, respectively). The quantification of the caffeic in red and white wines was also accomplished by standard addition method. The only sample treatment required in all cases consisted of an appropriate dilution with the supporting electrolyte solution, and the obtained results are in accordance with amounts reported in the literature for commercial samples [66,67].

An electrochemical sensor composed of Nafion–graphene nanocomposite film for the voltametric determination of caffeic acid (CA) has been proposed by Filik and co-workers [69]. A Nafion graphene oxide-modified glassy carbon electrode was fabricated by a simple drop-casting method and then graphene oxide was electrochemically reduced over the glassy carbon electrode. The electrochemical analysis method was based on the adsorption of caffeic acid on Nafion/ERGO/GCE and then the oxidation of CA during the stripping step. The resulting electrode showed an electrocatalytical response to the oxidation of caffeic acid (CA). At optimized test conditions, the calibration curve for CA showed two linear segments: The first linear segment increased from 0.1 to 1.5 μM and second linear segment increased up to10 μM. The detection limit was determined as 9.1 × 10^−8^ M using SWV. Finally, the proposed method was successfully used to determine CA in white wine samples.

An electrochemical sensor based on molecularly imprinted siloxanes (MIS) film was developed for the selective determination of CA by Kubota group [70]. The MIS film was prepared by sol-gel process, using the acid catalysed hydrolysis and condensation of tetraethoxysilane (TEOS), phenyltriethoxysilane (PTEOS), and 3-aminopropyltrimethoxysilane(3-APTMS) in the presence of CA as a template molecule. The MIS film was immobilized onto Au electrode surface pre-modified with 3-mercaptopropyltrimethoxysilane (3-MPTS). Under the optimized conditions, by using differential pulse voltammetry (DPV), the sensor showed a linear current response to the target CA concentration in the range from 0.500 to 60.0 μM, with a detection limit of 0.15 μM. The film exhibited high selectivity toward the template CA, as well as good stability and repeatability for CA determinations. Furthermore, the proposed sensor was applied to determine CA in wines samples and the results agreed with those obtained by a chromatographic method.

Curulli’s group developed a PEDOT (poly(3,4-ethylenedioxy) thiophene) modified Pt sensor for the determination of caffeic acid (CA) in wine [83]. Cyclic voltammetry (CV) with the additions standard method was used to quantify the analyte at PEDOT modified electrodes. PEDOT films were electrodeposited on platinum electrode (Pt) in aqueous medium by galvanostatic method using sodium poly(styrene-4-sulfonate) (PSS) as electrolyte and surfactant. CV allows for detecting the analyte over a wide concentration range (10.0 nM–6.5 mM) with a detection limit of 3.0 nM. The electrochemical method proposed was applied to determine CA in white and red wines and the obtained results are in accordance with amounts reported in literature for commercial samples [65,66].

Thangavelu [71] reported a caffeic acid (CA) electrochemical sensor using reduced graphene oxide and polydopamine composite modified glassy carbon electrode (RGO@PDA/GCE). Cyclic voltammetry (CV) was used to investigate the electrochemical behaviour of different modified electrodes (bare GCE, RGO/GCE and RGO@PDA/GCE) toward oxidation of CA and the CV results showed that RGO@PDA composite has higher electrocatalytic activity to related CA oxidation than other modified electrodes. Differential pulse voltammetry was used for determination of CA and the response of CA was linear over the concentration ranging from 5.0 nM to 450.55 μM with the low detection limit of 1.2 nM. The composite modified electrode has acceptable selectivity in the presence of interfering species. The practical applicability of the composite was evaluated in wine samples and the obtained recovery of CA in wine samples confirms its potential for practical applications.

An Au-PEDOT/reduced graphene oxide nanocomposite (Au-PEDOT/rGO) modified glassy carbon electrode (GCE) showed significantly high electrocatalytic activity toward caffeic acid (CA) oxidation as compared to bare GCE and Au-PEDOT modified electrode [72]. Under optimized conditions, the Au–PEDOT/rGO constructed sensors exhibited a wide linear range of 0.01–46 μM with a detection limit as low as 0.004 μM for the detection of CA. To validate its possible application, the present sensor showed a satisfactory anti-interference performance, high reproducibility, and sensitivity for the determination of CA in the red wine sample.

Liu et al. proposed Pd-Au/PEDOT/graphene (Pd-Au/PEDOT/rGO) nanocomposites, easily prepared in an aqueous medium by a one-pot method [73]. In addition to the Pd-Au nanoparticles deposited on the PEDOT polymer structure, dendritic bimetallic Pd–Au nanoclusters were also observed in the composites. The morphology of the nanocomposite was greatly influenced by the molar ratio of the added metal salt precursors. X-ray photoelectron spectroscopy and X-ray diffraction analyses indicated the presence of synergism and electron exchange between the Pd–Au bimetallic nanoclusters. The resultant nanocomposite improved the electronalytical parameters of the obtained sensor for CA detection. Under optimized conditions, the Pd-Au/PEDOT/rGO modified glassy carbon electrode showed a linear response range of 0.001–55 μM and a detection limit of 0.37 nM. The practical application of the Pd-Au/PEDOT/rGO/GCE was tested to determine CA in red wine samples and a good recovery was obtained. The proposed method proved sufficient for practical applications without any sample pretreatment.

A well-defined one-dimensional (1D) rod-like strontium vanadate (SrV_2_O_6_) was prepared by simple hydrothermal method without using any other surfactants/templates [74]. The formation of rod-like SrV_2_O_6_ was confirmed by various analytical and spectroscopic techniques. The as-prepared rod-like SrV_2_O_6_ was employed as material for assembling an electrochemical sensor for the detection of caffeic acid (CA) as well as visible light active photocatalyst for the degradation of metronidazole (MNZ) antibiotic drug. As an electrochemical sensor, the SrV_2_O_6_ modified glassy carbon electrode (GCE) showed an electrocatalytic activity for the detection of CA by chronoamperometry (CA) and cyclic voltammetry (CV). In addition, the electrochemical sensor showed good selectivity, a linear response range of 0.01–207 μM, and a detection limit of 4 nM.

Shen-Ming Chen reported the simultaneous electrochemical deposition of gold and palladium nanoparticles on graphene flakes (Au/PdNPs-GRF) for the sensitive electrochemical determination of caffeic acid (CA) [75]. The electrochemical determination of CA at Au/Pd NPs deposited on GRF were studied by using cyclic voltammetry and differential pulse voltammetry. Au/PdNPs-GRF electrode exhibited electrocatalytic activity towards CA with wide linear range of 0.03–938.97 μM and detection limit of 6 nM, respectively. Moreover, the Au/PdNPs-GRF was found to be a selective and stable active material for the sensing of CA. In addition, the proposed sensor was employed successfully in real sample analysis of fortified wines.

As a proof-of-concept, an eco-friendly Au@α-Fe_2_O_3_@RGO ternary nanocomposite modified glass carbon electrode (GCE) was investigated for electrochemical detection of caffeic acid [76]. Characterization studies confirmed Au and α-Fe_2_O_3_ nanoparticles are uniformly distributed on the surfaces of reduced graphene oxide (RGO) nanosheets. The electrochemical mechanism involves the synergistic electrocatalytic activity of Au and α-Fe_2_O_3_ towards caffeic acid oxidation, with the RGO serving as an efficient electron shuttling mediator enhancing the sensor performance. The Au@α-Fe_2_O_3_@RGO modified GCE caffeic acid sensor showed a linear response range of 19–1869 μM and a detection limit of 0.098 μM. This ternary nanocomposite displays catalytic performance as well as selectivity toward caffeic acid. To demonstrate the potential application of the Fe_2_O_3_@RGO modified GCE caffeic acid sensor, caffeic acid in a coffee sample was measured.

Pt-PEDOT/reduced graphene oxide (Pt-PEDOT/rGO) nanocomposites were easily prepared by a simple and cost-effective one-pot method and used for modifying GCE for CA detection. The Pt-PEDOT/rGO nanocomposites modified glassy carbon electrode (Pt-PEDOT/rGO/GCE) displayed higher charge transfer efficiency and electrocatalytic activity to the oxidation of caffeic acid (CA) in comparison with the bare GCE, and rGO modified electrodes (rGO/GCE). Furthermore, the Pt-PEDOT/rGO/GCE showed a linear range of 5.0 × 10^−9^ to 5.0 × 10^−5^ M with a low detection limit of 2.0 × 10^−9^ M for the CA determination. Finally, the modified electrode was also applied to detect CA in green tea and black tea samples with promising recovery results [77].

Pt/Cu nanocrystal with uniform trifurcate structure has been synthesized by using KI as structure-directing agent during the reduction process for developing an electrochemical sensor for CA detection [78]. Due to the large surface area of this synthesized dendritic trifurcate nanocrystal and the synergistic effect between Pt and Cu, this nanocatalyst showed interesting performances when applied to the electrochemical detection of caffeic acid. Specifically, the detection limit is 0.35 μM and the linear range is from 1.2–1.9 μM. Finally, the modified electrode was also applied to detect CA in red wine samples with good recovery data.

A nanocatalyst based on copper sulphide nanodots grown on graphene oxide sheets nanocomposite (Cu_2_SNDs@GOS NC) was synthesized by a simple sonochemical technique and applied in electrocatalytic sensing of caffeic acid (CA) [79]. The nanocomposite modified screen printed carbon electrode (SPCE) showed electrocatalytic performance towards CA oxidation. Under optimized conditions, Cu_2_S NDs@GOS NC modified electrode showed fast and sensitive amperometric responses towards CA. The linear range was 0.055–2455 μM and the detection limit is 0.22 nM. The Cu_2_S NDs@GOS NC/SPCE can quantify the amount of CA present in carbonated soft drinks and red wine without sample pretreatment.

Electrosynthesized PEDOT layers with different thickness and different doping counterions are investigated for their application in electrochemical CA sensing [80]. In terms of electroanalytical characteristics, adsorption control provides higher electroanalytical sensitivity in a narrow concentration range of linear response. Diffusion control results in markedly lower sensitivity values but extended range of linearity in the concentration dependence. The use of a non-linear calibration curve provides an extended concentration range for electroanalytical purpose under the established adsorption-controlled conditions. Doping counterions (polystyrene sulfonate and dodecyl sulphate) have no effect on the electroanalytical performance of PEDOT for CA oxidation.

An environmentally friendly sensor for the quantification of caffeic acid (CA) is reported [81]. Bismuth decorated multi-walled carbon nanotubes drop casted with cetyltrimethylammonium bromide demonstrated synergistic catalytic properties on enhancing the surface area of the carbon paste electrode. The proposed modified sensor was used to determine CA by differential pulse voltammetry (DPV) technique. The best results were obtained at physiological pH where the response was linear over a range of 6.0 × 10^−8^ to 5.0 × 10^−4^ M and with a limit of detection of 0.157 nM. The proposed sensor exhibited good performances vs. the most common interferents. The detection of CA in samples, such as coconut water, teas, and fruit juices without pretreatments was also reported.

Aicheng Chen reported a novel electrochemical sensor developed using fluorine-doped graphene oxide (F-GO) for the detection of caffeic acid (CA) [82]. The electrochemical behaviour of bare glassy carbon electrode (GCE), F-GO/GCE, and GO/GCE toward the oxidation of CA were studied using cyclic voltammetry (CV), and the results obtained from the CV investigation revealed that F-GO/GCE exhibited a high electrochemically active surface area and electrocatalytic activity. Differential pulse voltammetry (DPV) was employed for the CA detection, obtaining a linear concentration range from 0.5 to 100.0 μM with a limit of detection of 0.018 μM. Furthermore, the sensor selectivity was tested for the presence of other hydroxycinnamic acids and ascorbic acid. Moreover, the F-GO/GCE offered a good sensitivity, long-term stability, and a good reproducibility. The practical application of the electrochemical F-GO sensor was verified using several samples of commercially available wine. The developed electrochemical sensor can directly detect CA in wine samples without pretreatment, making it a promising candidate for food and beverage quality control.

#### 4.1.2. Chlorogenic Acid

Chlorogenic acid (CGA, 5-*O*-caffeoylquinic acid, chemical structure in Figure 3) is a naturally occurring phenolic compound metabolized by several plants such as *Calendula officinalis* and *Echinacea purpurea* [84,85].

Clinical investigations demonstrated that CGA can be classified as an anti-hypertension, anti-inflammatory, anti-carcinogenic, anti-bacterium, and antitumor agent [84]. Furthermore, there are also reports that CGA has a potential role in the regulation of glycemia levels in type 2 diabetes [84]. Therefore, the quantitative measurement of CGA in plant materials has attracted great interest, and different analytical methods have been employed for the determination of CGA, including high performance liquid chromatography, capillary electrophoresis, and electrochemical methods [85]. Nevertheless, electrochemical methods provide a rapid and simple procedure, relatively low-cost equipment, and possible in situ analysis. However, there are only few reports on the electrochemical behaviour/detection of CGA. Considering the electrochemical approach involving nanomaterials, examples are reported in Table 2.

Gao reported and described a multi-walled carbon nanotube modified screen-printed electrode (MWCNTs/SPE), employed for the electrochemical determination of CGA [86]. Cyclic voltammetry (CV) and differential pulse voltammetry (DPV) methods for the determination of CGA were proposed. Under the optimal conditions, the sensor exhibited linear ranges from 4.8 × 10^−^^4^–4.4 × 10^−^^2^ M, and the detection limit for CGA of 3.38 × 10^−^^4^ M. According to the proposed analytical method, the MWCNTs/SPE was applied to the determination of CGA in coffee beans and the recovery was 94.74–106.65%. The result of CGA determination was in good agreement with those obtained by HPLC.

The preparation of a modified carbon paste electrode using highly defective mesoporous carbon (DMC) and room temperature ionic liquid 1-Butyl-3-methylimidazolium hexafluorophosphate (BMIM.PF_6_) was described [85] and the obtained sensor was employed for the electrochemical detection of chlorogenic acid (CGA) in herbal extracts. DMC with defective structure was synthesized via a facile method using nanosilica as a hard template, sucrose as a carbon source, and KNO_3_ as defect inducing agent. The proposed sensor exhibited good performances toward the electrochemical reaction of CGA in aqueous solution. The electrocatalytic behaviour was further considered as a detection procedure for the CGA determination by SWV. Under optimized conditions, the linear response range and detection limit were 2.00 × 10^−8^–2.50 × 10^−6^ M and 1.00 × 10^−8^ M, respectively. The method was successfully applied for determination of CGA in extracts of *Calendula officinalis* and *Echinacea purpurea*.

A novel gold nanoparticles-doped TAPB-DMTP-COFs (TAPB, 1,3,5-tris(4-aminophenyl)benzene; DMTP, 2,5-dimethoxyterephaldehyde; COFs, covalent organic frameworks) composite was prepared via COFs as the host matrix to support the growth of gold nanoparticles. Then, this composite was used to assemble an electrochemical sensor and presented a good electrocatalytic activity toward the oxidation of chlorogenic acid (CGA) in the phosphate buffer solution (pH 7.0) [87]. This electrochemical sensor displays a linear range of 1.00 *×* 10^−8^–4.00 × 10^−5^ M, with a detection limit of 9.50 *×* 10^−9^ M. According to the proposed method, the sensor was applied to the determination of CGA in coffee, fruit juice, and in herbal extracts, and the recovery was 99.20–102.50%. The results of CGA determination were in good agreement with those obtained by HPLC.

An innovative nanocomposite of multiwalled carbon nanotubes (MWCNTs), copper oxide nanoparticles (CuONPs) and lignin (LGN) polymer were successfully synthesized and used to modify the glassy carbon electrode for the determination of chlorogenic acid (CGA). Cyclic voltammetry (CV) showed a quasi-reversible, adsorption-controlled and pH dependent behaviour. Differential pulse voltammetry (DPV) was applied and used for the quantitative detection of CGA. Under optimal conditions, the proposed sensor showed linear responses from 5 mM to 50 mM, whilst the limit of detection was found to be 0.0125 mM. The LGN-MWCNTs-CuONPs-GCE were applied to detect the CGA in real coffee samples with the recovery ranging from 97 to 106% [88].

The as synthesized ZnO@PEDOT:PSS modified glassy carbon electrode (GCE) was employed for the electrochemical detection of chlorogenic acid (CGA) [89]. The composite structure is formed by zinc oxide (ZnO) having the flower-like structure and by poly(3,4-ethylenedioxythiophene) polymer doped with poly(4-styrenesulfonate) (PEDOT: PSS). The nanocomposite was synthesized successfully through the one-pot chemical synthesis. The physicochemical properties of the synthesized composite were studied by using various analytical techniques. Specifically, ZnO@PEDOT:PSS-GCE towards CGA oxidation shows a linear range of 0.03–476.2 μM and limit of detection 0.02 μM, respectively. Furthermore, the ZnO@PEDOT:PSS-GCE effectiveness for the determination of CGA is tested by using coffee powder and soft drink as real samples.

The detection of chlorogenic acid in clinic samples for metabolic kinetics studies is a challenging task due to the low concentration and lack of sensitive analytical methods. Concerning this analytical issue, a porous pencil lead electrode (PLE) has been used to detect chlorogenic acid by SWV technique. The sensitivity was significantly improved due to the accumulation of CGA at the porous structure of the electrode. Under the optimized conditions, the concentration of chlorogenic acid was linear in the range of 7.7 × 10^−^^8^–7.7 × 10^−^^6^ M with detection limits of 4.5 × 10^−^^9^ M. The electrode was used for the determination of chlorogenic acid in human urine with near 100% recovery [90].

#### 4.1.3. Rosmarinic Acid

Rosmarinic acid (RA, chemical structure in Figure 4) is the ester of caffeic acid, predisposed to biological activities. [91]. It is also one of the natural antioxidants acting as an antiseptic, antiviral, antibacterial and anti-inflammatory agent. RA is an important and major secondary metabolite present in some plants and applied also in various fields such as medicine, cosmetics, and food industry. Due to the known impacts of RA on Alzheimer’s disease, it was used to improve cognitive function as well as to reduce the severity of the renal disease [91].

Several studies have been reported concerning the determination of RA in different matrices using different analytical approaches such as chromatography, fluorescence, electrochemical sensors (very few papers), capillary electrophoresis [92].

Two different electrochemical sensing approaches are reported below the electrochemical behaviour of rosmarinic acid at the surface of a DNA-coated electrode was investigated using square-wave stripping voltammetry [93]. The voltammetric studies showed that rosmarinic acid is oxidized in two successive pH-dependent steps, each involving the transfer of two electrons and two protons. These oxidations correspond to two electroactive catechol groups. Moreover, strong interaction between the immobilized DNA and rosmarinic acid accumulates RA on the electrode surface resulting in an efficient preconcentration leading to high sensitivity of the sensor for rosmarinic acid determination. Several experimental parameters affecting the sensor response were optimized. Under optimized conditions, a linear concentration range of 0.040–1.5 μM with a detection limit of 0.014 μM was obtained. The proposed method was applied to the analysis of a rosemary extract. The obtained data was in good agreement with those obtained from HPLC analysis.

A modified carbon paste electrode (CPE) was designed and fabricated used magnetic functionalized molecularly imprinted polymer (MMIP) nanostructure for selective determination of RA in some plant extracts [93]. The MMIP nanostructure functionalized with −NH_2_ group (Fe_3_O_4_@SiO_2_@NH_2_) was prepared by surface imprinting approach and then used as modifier for the sensor. The nano sized functionalized MMIP particles (i.e., Fe_3_O_4_@SiO_2_@NH_2_) superparamagnetic behaviour was analysed and verified. In addition, cyclic voltammetry (CV) and differential pulse voltammetry (DPV) techniques were used for study the electrochemical behaviour and determination of RA on the modified CPE. Based on the results, the modified sensor has good sensitivity and selectivity for the detection of RA in the presence of other species compared to unmodified electrodes. Under optimum conditions, two linear concentration ranges (0.1–100 μM and 100–500 μM) with a detection limit of 0.085 μM were obtained. Finally, the applicability of the designed sensor was examined for determination of RA in four plant extracts, including *Salvia officinalis*, *Zataria multiflora*, *Mentha longifolia*, and *Rosmarinus officinalis*.

### 4.2. Caffeine

Caffeine (CAF, chemical structure in Figure 5) a natural alkaloid, distributed in seeds, nuts, or leaves of a number of plants, mainly coffee, cocoa, tea, with the natural function as insecticide, is the most widely consumed psychoactive substance in human dietary. Many physiological effects of CAF are well known, from stimulation of the central nervous system, diuresis, and gastric acid secretion [94] to nausea, seizures, trembling, and nervousness. Mutation effects on DNA have been also reported. Moreover, it is considered a risk molecule for cardiovascular diseases. Recently, also an antioxidant activity has been suggested for CAF, showing protective effects against oxidative stress. The presence of CAF in many beverages and drug formulations of worldwide economic importance [94] makes it an analyte of great interest and although many different analytical methods are currently applied, novel analytical methods for fast, sensitive and reliable determination of CAF are always necessary, especially for particular purposes, as the determination in specific matrix in the presence of interfering agents, or in a specific concentration range, besides under beneficial conditions in terms of time consumption, material cost, and procedure ease [94].

Electrochemical determination of CAF on different electrode materials, usually bare and miscellaneously modified carbon-based electrodes, has been also reported in literature [94]. The electrochemical methods offer a series of practical advantages as procedure ease, not too expensive instruments, the possibility of miniaturization, besides good sensitivity, wide linear concentration range, suitability for real-time detection, and reduced sensitivity to matrix effects. Several examples, concerning CAF detection via nanomaterials based electrochemical sensors are reported in Table 3.

As a first example of nanomaterials application to detect caffeine, we would like to mention a voltammetric sensor for caffeine, based on a glassy carbon electrode modified with Nafion and graphene oxide (GO) [95]. It exhibits a good affinity for caffeine (resulting from the presence of Nafion), and good electrochemical response (resulting from the presence of GO) for the oxidation of caffeine. The electrode enables the determination of caffeine in the range from 4.0 × 10^−^^7^ to 8.0 × 10^−^^5^ M, with a detection limit of 2.0 × 10^−^^7^ M. The sensor displays good stability, reproducibility, and high sensitivity. It was successfully applied to the quantitative determination of caffeine in beverages. Kubota group described the development of a novel sensitive molecularly imprinted electrochemical sensor for the detection of caffeine [96]. The sensor was prepared on a glassy carbon electrode modified with multiwall carbon nanotubes (MWCNTs)/vinyltrimethoxysilane (VTMS) recovered by a molecularly imprinted siloxane (MIS) film prepared by sol–gel process. MWCNTs/VTMS was produced by a simple grafting of VTMS on MWCNTs surface by in situ free radical polymerization. The siloxane layer was obtained from the acid-catalyzed hydrolysis/condensation of a solution constituted by tetraethoxysilane (TEOS), methytrimethoxysilane (MTMS), 3-(aminopropyl)trimethoxysilane (APTMS), and caffeine as a template molecule. The MIS/MWCNTs-VTMS/GCE sensor was tested in a solution of the caffeine and other similar molecules. After optimization of the experimental conditions, the sensor showed a linear response range from 0.75 to 40 μM with a detection limit (LOD) of 0.22 μM. The imprinted sensor was successfully tested to detect caffeine in real samples.

Pumera group reported and discussed the electrochemical detection of caffeine at different chemically modified graphene (CMG) surfaces with different defects and oxygen functionalities [97]. The analytical performances of graphite oxide (GPO), graphene oxide (GO), and electrochemically reduced graphene oxide (ERGO) were compared for the detection of caffeine. It was found that ERGO showed the best analytical parameters, such as lower oxidation potential, sensitivity, linearity range (5.00 × 10^−^^5^–3.00 × 10^−^^4^ M) and reproducibility of the response. ERGO was then used for the analysis of real samples. Caffeine levels of soluble coffee, teas, and energy drinks were measured without the need of any sample pre-treatment.

A solution containing anisotropic gold nanoparticles (AuNPs), chitosan (CHIT), and ionic liquid (IL, i.e., 1-butyl-3-methylimidazolium tetrafluoroborate, (BMIM) (BF_4_)) was cast on a graphene(r-GO) modified glassy carbon electrode to assemble an electrochemical sensor for the determination of theophylline (TP) and caffeine (CAF) [98]. The factors influencing this analytical approach are investigated, including the ingredients of the hybrid film, the concentrations of r-GO, HAuCl_4_, and IL, and the buffer solution pH. Under the optimized conditions, the linear response ranges are 2.50 × 10^−^^8^–2.10 × 10^−^^6^ M and 2.50 × 10^−^^8^–2.49 × 10^−^^6^ M for TP and CAF, respectively. The detection limits are 1.32 × 10^−^^9^ M and 4.42 × 10^−^^9^ M, respectively. The electrochemical sensor shows good reproducibility, stability, and selectivity, and it has been applied to the determination of TP and CAF in real samples. A sensitive and selective nanocomposite imprinted electrochemical sensor for the determination of caffeine has been proposed by Rezaei [99]. The imprinted sensor was fabricated on the surface of pencil graphite electrode (PGE) via one-step electropolymerization of the imprinted conductive polymer, including gold nanoparticles (AuNPs), and caffeine. Because of the presence of specific binding sites on the molecularly imprinted polymer (MIP), the sensor responded quickly to caffeine. AuNPs were introduced for the enhancement of electrical response. The linear ranges of the MIP sensor were from 2.0 to 50.0 and 50.0 to 1000.0 nM, with the limit of detection of 0.9 nM. Furthermore, the proposed method was applied for the determination of caffeine in real samples (urine, plasma, tablet, green tea, energy, and soda drink).

Curulli group developed a simple and selective method for the determination of caffeine also in complex matrix at a gold electrode modified with gold nanoparticles (AuNPs) synthetized in a chitosan matrix in the presence of oxalic acid [94]. A scheme of the approach and method used for the caffeine detection is illustrated in Figure 5.

The electrochemical behaviour of caffeine at both gold bare and gold electrode modified with AuNPs was carried out in acidic medium by cyclic voltammetry (CV), differential pulse voltammetry (DPV) and electrochemical impedance spectroscopy (EIS). Electrochemical parameters were optimized to improve the electrochemical response to caffeine. The most satisfactory result was obtained using a gold electrode modified with AuNPs synthetized in a chitosan matrix in the presence of oxalic acid, in aqueous solution containing HClO_4_ 0.4 M as supporting electrolyte. The performance of the sensor was then evaluated in terms of linearity range (2.0 × 10^−^^6^–5.0 × 10^−^^2^ M), operational and storage stability, reproducibility, limit of detection (1.0 × 10^−^^6^ M) and response to a series of interfering compounds as ascorbic acid, citric acid, gallic acid, caffeic acid, ferulic acid, chlorogenic acid, glucose, catechin and epicatechin. The sensor was then successfully applied to determine the caffeine content in commercial beverages and the results were compared with those obtained with HPLC-PDA as an independent method and with those declared from manufacturers.

A novel graphene oxide-reduced glutathione modified carbon paste (GORGCP) sensor has been assembled for sensitive determination of CAF in 0.01 M phosphoric acid (H_3_PO_4_) solution of pH range (1.0–5.0) in both aqueous and surfactant media (0.5 mM sodium dodecylsulfate (SDS) [100]. The interaction of CAF over the surface of sensor was investigated using different electrochemical techniques. The linear detection range of CAF was between 8–800 μM with a limit of detection of 0.153 μM. The sensor showed a good selectivity and was applied for detecting CAF in real samples with recoveries, ranging from 97.76% to 101.36%.

Walcarius and co-workers [101] reported the more recent sensor for CAF detection involving nanomaterials. A nano-cobalt (II, III) oxide modified carbon paste (NCOMCP) sensor was prepared and applied to the determination of CAF in both aqueous (0.01 M H_2_SO_4_) and micellar media (0.5 mM sodium dodecylsulfate, SDS). The electrochemical behaviour of CAF was studied by means the most common electrochemical techniques. Using DPV, CAF was detected in a concentration range between 5.0 and 600 μM, with a limit of detection of 0.016 μM. The sensor showed good selectivity vs. the most common interferences and was applicable for CAF sensing in several commercial coffee samples with recoveries ranging from 98.9% to 101.9%.

### 4.3. Ascorbic Acid

L-Ascorbic acid (AA) or Vitamin C is a water-soluble vitamin and antioxidant, present in many biological systems and food. The biochemically and physiologically active form is the L-enantiomer, with a γ-lactone structure [102]. AA is an ideal scavenger free-radical and singlet oxygen or act as a chelating agent. As a strong antioxidant, it acts as a two-electron donor involving hydrogen atom transfer, giving rise to the ascorbate radical ion first, and finally to dehydroascorbic acid. The AA action prevents the oxidation of several compounds present in food and/or beverages. The deficiency of AA can cause several diseases, such as rheumatoid arthritis, Parkinson’s and Alzheimer’s diseases, and even cancer [102]. The excess of AA can result to several other diseases such as gastric irritation. Moreover, in the presence of heavy metal cations, the excess of AA has other drawbacks, because it can act as a pro-oxidant, in other words it limits its own antioxidant action tills to produce the reactive oxygen species, causing oxidative stress. Therefore, the determination of AA in biological fluids is very important for the diagnosis of such diseases. The quantitative determination of AA is also necessary for different application fields, including among others, cosmetics, drugs, and food [103].

Conventional bare electrodes, like Pt, Au, and glassy carbon, were used for the ascorbic acid detection but because of the vitamin C overpotential, fouling of the electrode surface was observed, involving low sensitivity and selectivity. Nanomaterials were used for the modification of conventional electrode to reduce the overpotential and to enhance the electrode sensitivity, selectivity, and reproducibility.

In Table 4, some interesting examples whereby the electrochemical sensors employing nanomaterials were designed and successfully applied for the AA detection in food analysis are reported. 

Yang and co-workers developed an electrochemical sensor for selective detection of ascorbic acid in the presence of dopamine (DA) and uric acid (UA), by modifying a glassy carbon electrode with carbon supported by NiCoO_2_ nanoparticles (NiCoO_2_/C) [104]. The sensor showed a linear concentration range from 1.0 × 10^−^^5^ M to 2.63 × 10^−^^3^ M, with a detection limit of 0.5 μM. The results highlighted that the NiCoO_2_ nanoparticles have improved the AA sensor performances, in comparison to a bare CGE (fast response time, high reproducibility, and stability). It was applied to the detection of AA in real samples, such as (fetal) bovine serum, vitamin C tablets, and soft drinks.

Khalilzadeh and co-workers reported a method for one-pot biosynthesis of silver nanoparticles (AgNPs) using onion extracts as reducing agent and stabilizer [105]. The synthesized AgNPs were characterized by spectroscopic and electrochemical methods. Finally, the AA detection was performed by means of square wave voltammetry and a linear concentration range was obtained from 0.4 to 450.0 μM. The detection limit for ascorbic acid was 0.1μM. The sensor was applied successfully to detect the AA content in several fruit juices.

A smartphone-based integrated voltammetry system using modified electrode was developed for simultaneous detection of biomolecules and it is a fine example to combine nanomaterials and innovative technological approach. The system included a disposable sensor, a coin-size detector, and a smartphone equipped with application program [113]. Screen-printed electrodes were used, where reduced graphene oxide and gold nanoparticles were electrochemically deposited. The smartphone is the core device to communicate with the detector, elaborate the data, and record voltammograms in real-time. Then, the system was applied to detect standard solutions of the biomolecules and their mixtures as examples. The results showed that the method allowed the selective detection of the target molecules. The system was then tested to detect the different molecules, including AA, in artificial urine and in beverages. with linear and high sensitivity.

Arduini and co-workers reported an interesting development of a carbon black nanomodified inkjet-printed sensor for ascorbic acid detection for AA detection [114]. The possibility to use the paper employed in printed electronics as a substrate to develop a paper-based sensor is analysed and evaluated. In Figure 6, the procedure to manufacture paper-based inkjet printed electrodes and the dimensions of the printed electrodes are shown.

To improve the electrochemical performances of the inkjet-printed sensor, a dispersion of carbon black nanoparticles was used to modify the working electrode, for assembling a nanomodified electrochemical sensor platform. This disposable sensor was characterized both electrochemically and morphologically, and it has been used to detect ascorbic acid as model analyte. It has been evidenced that the presence of carbon black as nano modifier decreased the overpotential for ascorbic acid oxidation in comparison to the unmodified sensor. Under optimized experimental conditions, this printed electrochemical sensor was successfully employed for the detection of ascorbic acid in a dietary supplement, and the results were in agreement with those declared from the manufacturers.

Rahman and co-coworkers modified with ZnO⋅CuO nanoleaves a flat GCE, fabricated through a conducting nafion polymer matrix in order to detect selective acetylcholine and ascorbic acid simultaneously [106].

Concerning ascorbic acid, a linearity range from 100.0 pM to 100.0 mM was obtained with a detection limit of 12.0 pM. ZnO·CuO nanoleaves were synthesized by using a wet-chemical technique. This sensor applied for the determination of acetylcholine and ascorbic acid in real samples, i.e., human, mouse, and rabbit serum, orange juice, and urine, with satisfactory results.

Decorated carbon nanoonions (CNOs) with nickel molybdate (NiMoO_4_) and manganese tungstate (MnWO_4_) nanocomposite were used to modify a glassy carbon electrode (GCE) and to detect AA in real samples [107].

The high conductivity and large surface area of the CNOs and the electrocatalytic effect of NiMoO_4_ and MnWO_4_ nanocomposite provide a synergistic effect on the electrooxidation processes of AA. The modified GCE surface with a synthesized nanocomposite of CNO-NiMoO_4_-MnWO_4_ was used to determine ascorbic acid using differential pulse voltammetry (DPV) method. The developed ascorbic acid sensor displayed a wide dynamic calibration range (1–100 μM) with a detection limit of 0.33 μM. The sensor was applied to determine AA in real samples of orange, tomato, strawberry, and pineapple juices with satisfactory recovery.

Vidrevich and co-workers developed a voltammetric sensor based on a carbon veil (CV) and phytosynthesized gold nanoparticles (Au-gr) for ascorbic acid (AA) determination [108].

Extract from strawberry leaves was used as source of antioxidants (reducers) for Au-gr phytosynthesis. The sensor exhibits a linear response to AA in a concentration range from 1 μM to 5.75 mM and a limit of detection of 0.05 μM. The sensor was successfully applied for the determination of AA content in fruit juices without samples preparation. A correlation (r = 0.9867) between the results of AA determination obtained on the developed sensor and integral antioxidant activity of fruit juices was observed.

A nonenzymatic amperometric sensor for determination ascorbic acid has been developed using the glassy carbon electrode (GCE) modified with mesoporous CuCo_2_O_4_ rods [109].

The mesoporous CuCo_2_O_4_ rods were prepared by hydrothermal method. The sample is pure spinel CuCo_2_O_4_ rods with mesoporous structure. The mesoporous CuCo_2_O_4_ rods showed a linearity range of AA concentration (1–1000 mM) with a linearity range from 1.00 to 1000 μM. The detection limit is down to 0.21 μM. The CuCo_2_O_4_/GCE AA sensor exhibited good performance in terms of stability, repeatability and selectivity and was appled to tetect AA in vitamin C tablets.

A simple, sensitive and precise electroanalytical method was developed using flow injection analysis (FIA) with amperometric detection and reduced graphene oxide sensor for ascorbic acid determination in samples of multivitamin beverages, milk, fermented milk, and milk chocolate [110].

No interference of sample matrix was observed, avoiding solvent extraction procedures (samples were only diluted). The FIA allowed a detection limit of 4.7 μM. Good precision (RSD < 7%) and accuracy (recoveries between 91 and 108%) evidenced the robustness of the method. The method was compared with ultra-fast liquid chromatography (UFLC), obtaining statistically similar results (95% confidence level).

A Cu-based nanosheet metal–organic framework (MOF), HKUST-1, was synthesised using a solvent method at room temperature. An indium tin oxide (ITO) electrode) was modified with this material to catalyse the electrochemical oxidation of ascorbic acid [111].

Under optimal conditions, the oxidation peak current at +0.02 V displayed a linear relationship with the concentration of AA within the ranges of 0.01–25 and 25–265 μM, respectively. The limit of detection was 3 μM. The porous nanosheet structure of HKUST-1 could explain the significative enhancement of the effective surface area and the electron transfer capaability. Moreover, the novel AA sensor demonstrated good reproducibility, stability, and high selectivity towards glucose, uric acid (UA), dopamine (DA), and several amino acids. It was also successfully applied to the real sample testing of various AA containing tablets.

Metal nanoclusters (NCs) are highly desirable as active catalysts due to their highly active surface atoms. Among the reported metal clusters, nickel nanoclusters (Ni NCs) have been less well developed than others, such as gold, silver and copper. A simple method was developed by Chen to synthesize atomically precise Ni clusters with the molecular formula of Ni_6_(C_12_H_25_S)_12_ [112].

The Ni_6_ NCs can be self-assembled into nanosheets due to their uniform size. It was found that the Ni_6_(C_12_H_25_S)_12_ clusters were loaded on carbon black and the resulting nanocomposite (Ni_6_ NCs/CB) was used to modify a GCE. The modified GCE was used to determine AA. The Ni_6_ NCs/CB based sensor showed high sensing performance for AA with a wide linear range (1–3212 μM) and a detection limit of 0.1 μM. The high catalytic activity could be due to the high fraction of surface Ni atoms with low coordination in the sub-nanometer clusters. Finally, Ni clusters can be used as highly efficient catalysts for the electrochemical detection of AA in tablets.

### 4.4. Nitrite

Nitrite is an additive used to extend the shelf-life of beverages and foods such as ham, salami, and other cured meats [9]. On the other hand, it proves harmful for the human body if it is added to food and beverages at levels higher than those indicated by the safety standards. The World Health Organization (WHO) has established a concentration limit of 3.0 mg·L^−1^ (65.22 μM) for nitrite in drinking water [115]. Therefore, the detection and quantification of nitrite is one of the critical issues of food analysis. Novel nanomaterials have been synthesized and used for the design of advanced sensors for the nitrite detection. It has been shown that nanomaterial-based electrocatalysts significantly improve analytical performances for the nitrite determination and several examples are reported in Table 5. We must evidence that the majority of them are concerned with the detection in water samples, but very significant examples are concerned with cured meats and milk.

As first example, we present a sensor based on silver nanoparticles stabilized by polyamidoamine (PAMAM) dendrimer, synthesized by a simple chemical reduction method [116]. The Ag-PAMAM modified electrode exhibited good electrocatalytic performance for the oxidation of nitrite. Under optimized conditions, in a pH 6.0 phosphate buffer solution, the sensor displayed a linear range from 4.0 μM to 1.44 mM with a detection limit of 0.4 μM. Furthermore, the proposed sensor was used for the determination of nitrite in tap water and milk samples.

Du and co-workers have prepared a promising electrochemical sensor based on Pt nanoparticles synthesized by a one-pot hydrothermal method and distributed on the surface of reduced graphene oxide (RGO) sheets [117]. The morphology and composition of as-prepared PtNPs-RGO composites have been characterized by different spectroscopic methods. The Pt-RGO modified electrode shows a good linearity between the peak current and concentration of nitrite (the range is not declared) with a detection limit of 0.1 μM. Compared to the Pt nanoparticles or the RGO modified glassy carbon electrode, the nanocomposite-based sensor shows a good reproducibility, stability, and anti-interference electrocatalytic performance toward nitrite sensing. Finally, it has been applied to detect nitrite in beverages.

Cellulose is the most abundant, renewable, biodegradable, natural polymer resource on earth and can be a good substrate for catalysis. For this reason, Zhang and co-workers used straw cellulose, oxidized by 2,2,6,6 tetra-methylpiperidine-1-oxylradical (TEMPO), and then they synthesized via a hydrothermal method a TEMPO oxidized straw cellulose/molybdenum sulphide (TOSC-MoS_2_) composite for modifying a GCE [118]. The TOSC-MoS_2_ modified glassy carbon electrode is used as electrochemical sensor for detecting nitrite. Cyclic Voltammetry (CV) results showed TOSC-MoS_2_ has electrocatalytic activity for the oxidation of nitrite. The amperometric response results indicate the TOSC-MoS_2_ modified GCE can be used to determine nitrite with linear ranges of 6.0–3140 and 3140–4200 μM and a detection limit of 2.0 μM. The proposed sensor has good anti-interference performance, and it has been applied to real sample analysis (river and drinking water).

Shen-Ming Chen group has prepared and evaluated a palladium nanoparticles (PdNPs) decorated functionalized multiwalled carbon nanotubes (f-MWCNT) modified glassy carbon electrode for the amperometric determination of nitrite in different water samples [119]. The f-MWCNT/PdNPs composite modified electrode was prepared by electrodeposition of PdNPs on the surface of f-MWCNT. The resultant sensor exhibits excellent electrocatalytic activity towards the oxidation of nitrite compared to MWCNT, f-MWCNT, and PdNPs modified electrodes. The amperometric method was used to determine nitrite and the response of the nitrite on modified electrode was linear in the concentration range from 0.05 to 2887.6 μM with a detection limit of 22 nM. Sensor was further applied to detect nitrite in different water samples (river, pond, and drinking water).

A highly sensitive electrochemical method was developed by Kalcher and co-workers for the determination of nitrites in tap water using a glassy carbon electrode modified with graphene nanoribbons (GNs/GCE) [120]. Graphene nanoribbons (GNs) have been synthetized and aligned to the surface of glassy carbon electrode (GCE) and showed good electrocatalytic activity for nitrite oxidation. Studies about electrochemical behaviour and optimization of the most important experimental conditions were performed using cyclic voltammetry (CV) in Britton–Robinson buffer solution (BRBS), while quantitative studies were undertaken with amperometry. The influence of most common interferents was found to be negligible. Under optimized experimental conditions linear calibration curves were obtained in the range from 0.5 to 105 μM with the detection limit of 0.22 μM. The proposed method and sensor are successfully applied for the determination of nitrite present tap water samples without any pretreatment.

A green route was used to synthesize Pd/Fe_3_O_4_/polyDOPA/RGO composite. The in-situ nucleation and growth of Fe_3_O_4_ and Pd nanostructures was performed on reduced graphene oxide (RGO), based on poly DOPA (3,4-Dihydroxy-l-phenylalanine, DOPA) [121]. This composite showed an electrocatalytic activity toward nitrite oxidation. The amperometric results indicated that Pd/Fe_3_O_4_/poly DOPA/RGO modified glassy carbon electrode was employed to determine nitrite with a linear range of 2.5–6470 μM and a detection limit of 0.5 μM. With good anti-interference behaviour and good stability, the proposed sensor was successfully applied for the determination of nitrite in Yellow River water and sausage extract with satisfactory results.

Flexible and free-standing reduced graphene oxide (rGO) papers, doped with dye molecules such as phenazine, phenothiazine, phenoxazine, xanthene, acridine, and thiazole, were used to assemble an electrochemical sensor for nitrite detection by Aanyalıoglu [122]. The assembling procedure included two basic steps: vacuum filtration of a dispersion containing GO sheets and dye molecule through a membrane and reduction of free-standing GO/dye paper to rGO/dye paper by treatment with hydriodic acid. Electrical conductivity and electrochemical performance studies indicated that acriflavine (Acr) doped rGO paper electrode showed the best performance for the oxidation of nitrite. The resulted amperometric sensor was very stable, flexible, and reproducible, with a detection limit of 0.12 μM and linear range of 0.40–3600 μM. Under optimized conditions, it was applied to detect nitrite in milk as well as mineral and tap water samples.

A screen-printed amperometric sensor based on a carbon ink bulk-modified with graphene oxide decorated with MnO_2_ nanoparticles (MnO_2_/GO-SPE) was prepared as a sensor for nitrite detection by Banks and co-workers [123]. The MnO_2_/GO-SPE showed an electrocatalytic activity for the electrochemical determination of nitrite in 0.1 M phosphate buffer solution (pH 7.4), with a limit of detection of 0.09 μM and with two linear ranges of 0.1–1 μM and 1–1000 μM, respectively. Additionally, the nitrite sensor presented a good selectivity, reproducibility, and stability and was applied to detect nitrite in tap and mineral water samples.

A paper-based, low-cost, disposable electrochemical sensing platform was developed for nitrite analysis based on graphene nanosheets and gold nanoparticles by Miao [132]. In comparison with the electrochemical responses at bare gold and glassy carbon electrodes, a considerably higher oxidation current was recorded. At the paper-based electrode, the fouling effect due to the oxidation products adsorption was negligible. Moreover, this paper-based sensing platform was applied to determine nitrite in environmental and food samples.

Glassy carbon and platinum electrodes were modified with composite films including gold nanoparticles (AuNPs) and a carbosilane-dendrimer with peripheral electronically communicated ferrocenyl units (Dend), by Alonso and co-workers [124]. The modified electrodes exhibited interesting electrocatalytic activity toward the oxidation of nitrite, giving higher peak currents at lower oxidation potentials than those at the bare electrode and at the dendrimer-modified electrode. Under the optimized conditions the sensor has a linear response in the 10 μM to 5 mM concentration range, and a detection limit of 2.0 × 10^−7^ M. The sensor was successfully applied to the determination of nitrite in natural water samples.

An electrochemical sensor based on Cu metal nanoparticles distributed on multiwall carbon nanotubes-reduced graphene oxide nanosheets (Cu/MWCNT/RGO) for determination of nitrite and nitrate ions was assembled by Rezaei group [125]. The morphology of the nanocomposite was investigated using different spectroscopic methods. Under optimized experimental conditions, the modified GCE showed catalytic activity toward the oxidation of nitrite with a significant increase in oxidation peak currents in comparison with those at bare GCE. Using SWV the sensor showed a linear concentration range from 0.1 to 75 μM with detection limit of 30 nM for nitrite. Furthermore, the sensor was applied to the detection of nitrite in the tap and mineral waters, sausages, salami, and cheese samples.

Highly ordered MOFs-derived α-Fe_2_O_3_/CNTs hybrids through in-situ insertion of carbon nanotubes and their subsequent calcination were prepared for the electrochemical detection of nitrite. Metal organic frameworks (MOFs), composed of metal ions and organic ligands connected each other through strong coordination bonds, have been widely applied in gas storage and separation [0]. The in-situ insertion of CNTs improves the electron-transfer between the α-Fe_2_O_3_ and CNTs, making α-Fe_2_O_3_ highly active for oxidation of nitrite. The sensor demonstrated good performances of nitrite detection with the limit of detection of 0.15 μM, and the linear range from 0.5 μM to 4000 μM. Finally, it was applied for the detection of nitrite in tap and pond water.

Another MOFs was used to prepare a sensor for detecting nitrite by Zhou and co-workers [127]. The MOFs were calcined in argon atmosphere to prepare magnetic Co nanocages (CoCN). The synthesized CoCN were deposited on the surface of glassy carbon electrode obtaining a magnetic sensing platfor(CoCN/MGCE). The prepared CoCN/MGCE displayed interesting performance for electrocatalytic oxidation of nitrite, with a linear concentration range of 5–705 μM with a 0.18 μM detection limit. The proposed sensor was applied for detecting nitrite in real samples of tap water. A glassy carbon electrode modified with Ag/Cu nanoclusters and multiwalled carbon nanotubes to detect nitrite was prepared using the electrodeposition method [128]. This electrode showed electrocatalytic activity for the oxidation of nitrite. Upon an increase of the nitrite concentration from 1.0 μM to 1.0 mM, the current response increased linearly, with a detection limit of 2 × 10^−7^ M. The electrode exhibited good stability, and it was applied in the detection of nitrite in lake water, drinking water and seawater.

An electrochemical sensor based on cobalt oxide decorated reduced graphene oxide and carbon nanotubes (Co_3_O_4_-rGO/CNTs) has been prepared for the nitrite detection. The sensor showed a linear concentration range from 8 μM to 56 mM and a detection limit of 0.016 μM. It was also applied to determine the nitrite level in tap water real samples with satisfactory recovery [129].

Core-shell Ni@Pt nanoparticles were synthesised on graphene [Ni@Pt/Gr] by Medhany [130]. This synthesized nanocomposite was characterized by different spectroscopic methods. Ni@Pt/Gr electrode showed electrocatalytic activity towards nitrite oxidation. Linear calibration curves were recorded at Ni@Pt/Gr electrodes obtaining a linear concentration range of 10–15 mM. The prepared nanocomposite could successfully estimate nitrite concentration in drinking water samples. Finally, another example of a sensor using a MOF system has been added. An electrochemical sensing platform based on a Cu-based metal-organic framework (Cu-MOF) decorated with gold nanoparticles (AuNPs) was assembled by Chen for the detection of nitrite [131]. AuNPs were electrodeposited on Cu-MOF modified glassy carbon electrode (Cu-MOF/GCE) using the potentiostatic method. The AuNPs decorated Cu-MOF (Cu-MOF/Au) displayed a catalytic effect for the oxidation of nitrite due to the high surface area and porosity of Cu-MOF, preventing the aggregation of AuNPs. The amperometry was adopted for quantitative determination of nitrite. The prepared electrochemical sensing platform demonstrates high sensitivity, selectivity, and good stability for the detection of nitrite. It shows two wide linear ranges of 0.1–4000 and 4000–10,000 μM, and a low detection limit of 82 nM. Moreover, the sensing platform can also be used for the nitrite detection in river water samples.

## 5. Conclusions

An overview of the most recent applications of nanomaterials and electrochemical sensors in food analysis has been presented.

It is evident that a plethora of nanomaterials including hybrid nanocomposites has been prepared, characterized, and used for different sensing applications, in many cases, the design has involved an accurate and targeted control of shape and dimensions of the nanostructure.

In many cases smart nanomaterials/nanocomposites, based on carbon nanostructures, metal nanoparticles/clusters and polymer both natural and/or synthetic, have been prepared and coupled with the most common electroanalytical techniques with very interesting results in terms of sensitivity, linearity range amplitude and selectivity. A challenge for the future is the application of all these promising results for defining widespread analytical protocols for addressing the food security issue, employing low cost, biocompatible materials and polymers as well as greener synthetic approaches and devices, e.g., screen printed electrodes, where the criticalities due to fouling are eliminated.

Although the majority the described sensors demonstrated a possible usefulness for food analysis area, they have been validated only in the laboratory. Generally, there are many electrochemical sensors for the food analysis of additives and contaminants. However, the number of the corresponding commercially available sensors is very limited. Precise and accurate validation studies are strongly suggested for a real employment of these electrochemical sensors for food and beverage safety testing. In addition, the impacts of environmental constraints, storage, selectivity, the matrix effects due to the complexity of food samples, and stability under real operative conditions must be verified. The validation and testing of statistically relevant numbers of samples, comparability, and interlaboratory studies to validate the robustness of such sensing platforms are the next critical steps for the achievement of industry level acceptance and regulatory approvals. It would be highly beneficial if these sensors were to be introduced by the food industry to monitor the safety and the quality of processed food and beverages.

Moreover, studies of the toxicity and degradation of these nanomaterials are required. All these issues should be further addressed before the introduction to the sensors market.

Finally, the development of smart sensors is linked to the development of portable devices. Improved portability may be achieved through the connectivity and integration of electrochemical sensors with devices such as smartphones and tablets, but very few examples are available [113]. Such an integration of two distinct areas of research (sensors and ICT), addressing the day-to-day needs of people, could facilitate the introduction of the next generation of smart sensors into the food processing industries to increase the quality and safety of food and beverages.

## Figures and Tables

**Figure 1 molecules-25-05759-f001:**
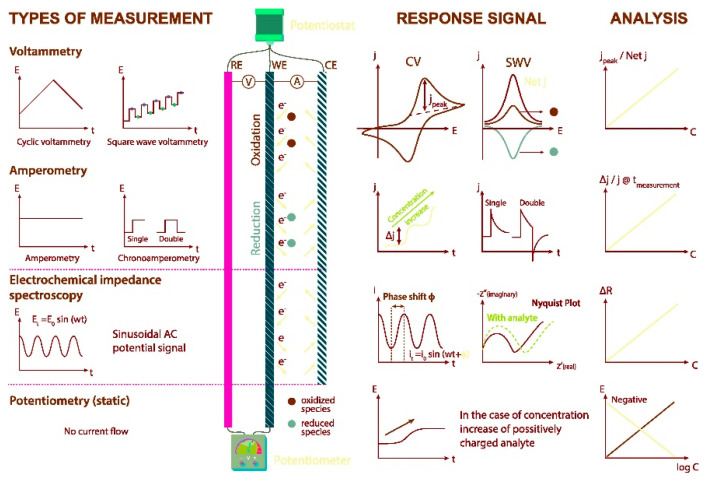
Overview of electrochemical methods of analysis: voltammetry, amperometry, electrochemical impedance spectroscopy (EIS), and potentiometry [19].

**Figure 2 molecules-25-05759-f002:**
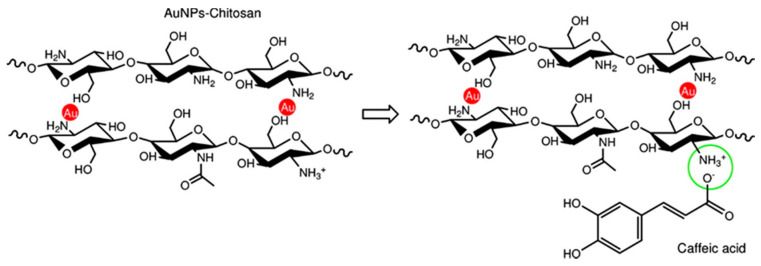
Schematic representation of the interaction between AuNPs/chitosan nanocomposite and caffeic acid. Reprinted with permission from [66] Copyright 2012, American Chemical Society.

**Figure 3 molecules-25-05759-f003:**
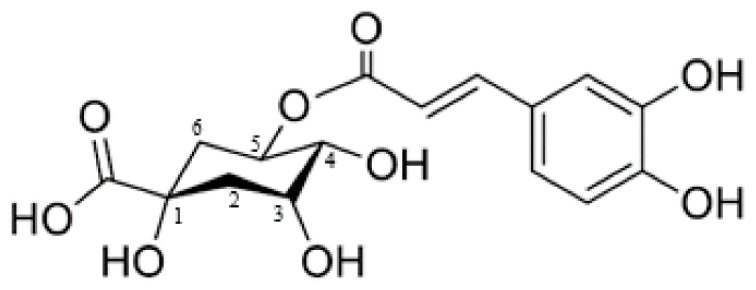
Chemical structure of chlorogenic acid [84].

**Figure 4 molecules-25-05759-f004:**
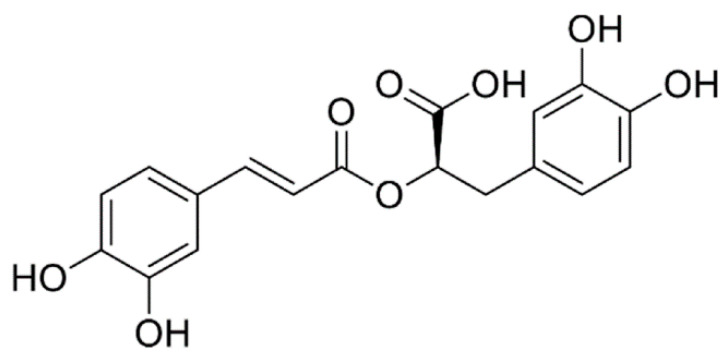
Chemical structure of rosmarinic acid [91].

**Figure 5 molecules-25-05759-f005:**
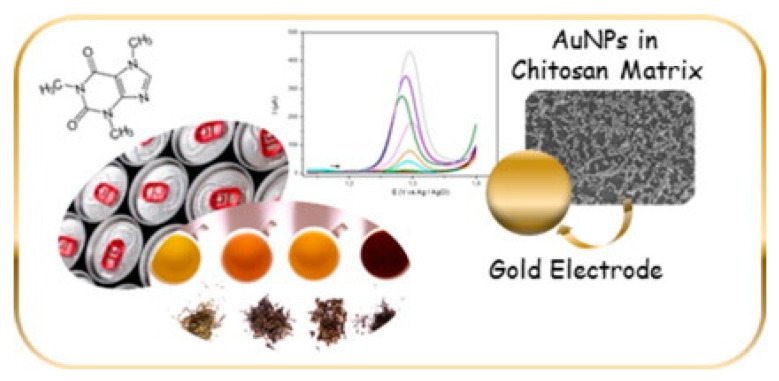
Scheme of the approach and method used for the caffeine detection reprinted with the permission from [94]. Copyright 2017 Elsevier.

**Figure 6 molecules-25-05759-f006:**
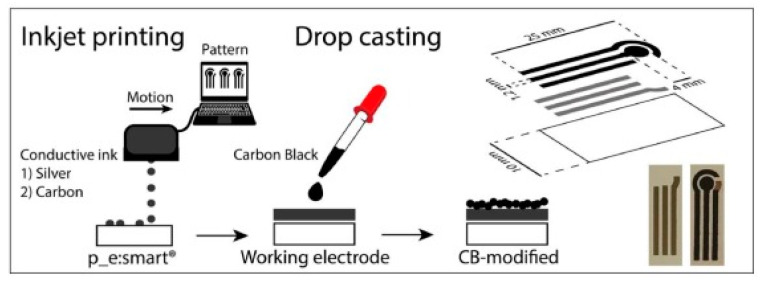
Procedure to manufacture paper-based inkjet printed electrodes and dimensions of the printed electrodes Reprinted with permission from [114]. Copyright 2018, Elsevier.

**Table 1 molecules-25-05759-t001:** An overview of recent electrochemical sensors for CA determination, using nanomaterials and/or nanocomposites.

Electrochemical Methods	Electrode Material	Linearity Range (mol·L^−1^)	LOD (mol·L ^−1^)	Application	Reference
DPV)	AuNps/Chitosan/AuE	5.00 × 10^−8^–2.00 × 10^−3^	2.50 ×10^−8^	Red and white wines	[66,67]
SWS	Nafion/ER-GO/GCE	1.0 × 10^−7^–1.0 × 10^−6^	9.1 × 10^−8^	White wines	[69]
DPV	MIS/AuE	5.00 × 10^−7^–6.00 × 10^−5^	1.50 × 10^−7^	White wines	[70]
DPV	RGO@PDA/GCE	5.0 × 10^−9^–4.55 × 10^−4^	1.20 × 10^−9^	Wines	[71]
DPV	Au–PEDOT/rGO/GCE	1.00 × 10^−8^–4.60 × 10^−5^	4.00 × 10^−9^	Red wines	[72]
DPV	PdAu/PEDOT/rGO/GCE	1.90 × 10^−9^–5.50 × 10^−5^	3.70 × 10^−10^	Red wines	[73]
Amperometry	SrV_2_O_6_/GCE	1.00 × 10^−8^–2.07 × 10^−4^	4.00 × 10^−9^	No real samples	[74]
DPV	Au/PdNPs/GRF/GCE	3.00 × 10^−8^–9.40 × 10^−4^	6.00 × 10^−9^	Fortified wines	[75]
DPV	Au@α-Fe_2_O_3_/RGO/GCE	1.90 × 10^−5^–1.87 ×10^−3^	9.80 × 10^−8^	Coffee samples	[76]
DPV	PEDOT/rGO/PtE	5.0 × 10^−9^–5.0 × 10^−5^	2.0 × 10^−9^	Teas	[77]
DPV	PtCu trifurcate nanocrystal/GCE	1.20 × 10^−6^–1.90 × 10^−3^	3.50 × 10^−7^	Red wines	[78]
Amperometry	Cu_2_S NDs@GOS NC/SPCE	5.50 × 10^−8^–2.50 × 10^−3^	2.20 × 10^−10^	Soft drinks and red wines	[79]
DPV	PEDOT/GCE	thin film 1.50 × 10^−7^–4.00 × 10^−6^ thick fim1.50 × 10^−6^–4.75 × 10^−5^		No real samples	[80]
DPV	MWCNTs-Bi/CTABCPE	6.0 × 10^−8^–5.0 × 10^−4^	1.91 × 10^−9^	Coconut water, teas, and fruit juices	[81]
DPV	F-GO/GCE	5.00 × 10^−7^–1.00 × 10^−4^	1.80 × 10^−8^	Red wines	[82]

**Table 2 molecules-25-05759-t002:** An overview of recent electrochemical sensors for CGA determination, using nanomaterials and/or nanocomposites.

Electrochemical Methods	Electrode Material	Linearity Range (mol·L^−1^)	LOD (mol·L^−1^)	Application	Reference
DPV	MWCNTs/SPE	4.8 × 10^−4^–4.4 × 10^−^^2^	3.38 × 10^−^^4^	Coffee beans	[86]
DPV	DMC/BMIM.PF_6_/CPE	2.00 × 10^−8^–2.50 × 10^−6^	1.00 × 10^−8^	Herbal extracts of Calendula officinalis and Echinacea purpurea	[85]
DPV	AuNps@TAPB-DMTP-COFs/GCE	1.00 × 10^−8^–4.00 × 10^−5^	9.50 × 10^−9^	Coffee, fruit juice and herbal extracts	[87]
Differential Pulse Voltammetry (DPV) WE	MWCNTs/CuONPs/LGN/GCE	5.00 × 10^−3^–5.00 × 10^−2^	1.25 × 10^−5^	Coffee	[88]
Differential Pulse Voltammetry (DPV) WE)	ZnO@PEDOT:PSS/GCE	3.00 × 10^−8^–4.76 × 10^−4^	2.00 × 10^−8^	Coffee powder, soft drink	[89]

**Table 3 molecules-25-05759-t003:** An overview of electrochemical sensors for CAF determination, using nanomaterials and/or nanocomposites.

Electrochemical Methods	Electrode Material	Linearity Range (mol·L^−1^)	LOD (mol·L^−1^)	Application	Reference
DPV	Nafion/GO/GCE	4.00 × 10^−7^–8.00 × 10^−5^	2.00 × 10^−7^	Soft and energy drinks, cola beverage	[95]
DPV	MIS/MWCNTs/VTMS/GCE	7.50 × 10^−7^–4.00 × 10^−5^	2.20 × 10^−7^	Coffees, energy drinks,	[96]
DPV	ERGO/GCE	5.00 × 10^−5^–3.00 × 10^−4^	Not declared	Cola beverage, tea, and soluble coffee	[97]
DPV	AuNps/chitosan–ionic liquid/Gr/GCE	2.50 × 10^−8^–2.49 × 10^−6^	4.42 × 10^−9^	Energy drink, teas, drugs	[98]
DPV	AuNps@PPY/PGE	2.00 × 10^−9^–5.00 × 10^−8^ 5.00 × 10^−8^–1.00 × 10^−6^	9.00 × 10^−10^	Soft and energy drinks, green tea, human plasma, drugs and urine	[99]
DPV	AuNps/chitosan/AuE	2.00 × 10^−6^–5.00 × 10^−2^	1.00 × 10^−6^	Cola beverages, energy drink, teas	[94]
DPV	GO/RG/CPE	8.00 × 10^−6^–8.00 × 10^−4^	1.53 × 10^−7^	Cola beverages, energy drink, teas, and drugs	[100]
DPV	CoON/CPE	5.00 × 10^−6^–6.00 × 10^−4^	1.60 × 10^−8^	Coffees	[101]

**Table 4 molecules-25-05759-t004:** An overview of electrochemical sensors for AA determination, using nanomaterials and/or nanocomposites.

Electrochemical Methods	Electrode Material	Linearity Range (mol·L^−1^)	LOD (mol·L^−1^)	Application	Reference
DPV	NiCoO_2_/C/GCE	1.00 × 10^−5^–2.63 × 10^−3^	5.00 × 10^−7^	Fetal bovine serum, Vitamin C tableys, Vitamin C drinks	[104]
SWV	AgNPs@onion extracts/CPE	4.00 x 10^−7^–4.50 × 10^−4^	1.00 × 10^−7^	Orange, kiwi and apple juices	[105]
DPV	ZnO⋅CuO NLs/GCE	1.00 × 10^−7^–1.00 × 10^−1^	1.20 × 10^−8^	Human, mouse, and rabbit serum, orange juice, and urine	[106]
DPV	CNO-NiMoO_4_-MnWO_4_/GCE	1.00 × 10^−6^–1.00 × 10^−4^	3.30 × 10^−7^	Orange, strawberry, tomato, pineapple juices	[107]
Amperometry	Mesoporous CuCo_2_O_4_/GCE	1.00 × 10^−4^–1. 05 × 10^−3^	2.10 × 10^−7^	Vitamin C tablets, Vitamin C effeverscent tablets and urine	[108]
LSV	Au-gr/CVE	1.00 × 10^−6^–5.75 × 10^−3^	5.00 × 10^−8^	Cherry-apple juice, apple juice for children, apple juice and apple nectar clarified	[109]
FIA-Amperometry	rGO/GCE	Linearity range not declared	4.70 × 10^−6^	Milk, fermented milk, chocolate milk and multivitamin supplement	[110]
DPV	HKUST-1/ITO	1.00 × 10^−5^–2. 65 × 10^−3^	3.00 × 10^−6^	Vitamin C pills, Vitamin C tablets, Vitamin C effervescent tablets	[111]
Amperometry	Ni_6_ NCs/CB/GCE	1.00 × 10^−6^–3.21 × 10^−3^	1.00 × 10^−7^	Vitamin C tablets	[112]

**Table 5 molecules-25-05759-t005:** An overview of electrochemical sensors for nitrite determination, using nanomaterials and/or nanocomposites.

Electrochemical Methods	Electrode Material	Linearity Range (mol·L^−1^)	LOD (mol·L^−1^)	Application	Reference
DPV	AgNPs@PAMAM/GCE	4.00 × 10^−6^–1.44 × 10^−3^	4.00 × 10^−7^	Milk and tap water	[116]
DPV	PtNPs/rGO/GCE	Linearity range not declared	1.00 × 10^−7^	Beverages	[117]
DPV	TOSC-MoS_2_/GCE	6.00 × 10^−6^–4.20 × 10^−3^	2.00 × 10^−6^	River and drinking water	[118]
Amperometry	f-MWCNT/PdNPs/GCE	5.00 × 10^−8^–3.00 × 10^−6^	2.20 × 10^−8^	River, pond, and drinking water	[119]
DPV	GNs/GCE	1.00 × 10^−6^ –1. 05 × 10^−4^	2.20 × 10^−7^	Tap water	[120]
DPV	Pd/Fe_3_O_4_/polyDOPA/RGO/GCE	2.50 × 10^−6^–6.47 × 10^−3^	5.00 × 10^−7^	River water and sausage	[121]
Amperometry	rGO/Acr/GCE	4.00 × 10^−7^–3.60 × 10^−3^	1.20 × 10^−7^	Milk, mineral and tap water	[122]
Amperometry	MnO_2_/GO-SPE	1.00 × 10^−7^–1.00 × 10^−3^	9.00 × 10^−8^	Tap and mineral water	[123]
DPV	AuNPs/carbosilane-dendrimer/GCE	1.00 × 10^−5^–5. 00 × 10^−3^	2.00 × 10^−7^	Natural water	[124]
SWV	Cu/MWCNT/RGO/GCE	1.00 × 10^−7^–7.50 × 10^−5^	3.00 × 10^−8^	Tap and mineral waters, sausages, salami, and cheese	[125]
DPV	MOFs-derived α-Fe_2_O_3_/CNTs/GCE	1.00 × 10^−7^–7.50 × 10^−5^	3.00 × 10^−8^	Tap and mineral waters, sausages, salami, and cheese	[126]
DPV	CoCNM/GCE	5.00 × 10^−6^–7.05 × 10^−6^	1.80 × 10^−7^	Tap water	[127]
DPV	Ag/CuNCs/MWCNTs/GCE	1.00 × 10^−6^–1.00 × 10^36^	2.00 × 10^−7^	Lake water, drinking water and seawater.	[128]
DPV	Co_3_O_4_@rGO/CNTs/GCE	8.00 × 10^−6^–5.60 × 10^−2^	1.60 × 10^−8^	Tap water	[129]
DPV	Ni@Pt/Gr/GCE	1.00 × 10^−5^–1.50 × 10^−2^	Not declared	Tap water	[130]
DPV	AuNPs@Cu-MOF/GCE	1.00 × 10^−7^–1.00 × 10^−2^	8.20 × 10^−8^	River water	[131]

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
