# Peer review of "Nanomaterials in Electrochemical Sensing Area: Applications and Challenges in Food Analysis"

_molecules, 2020, doi:10.3390/molecules25235759_

Round 1

Reviewer 1 Report

Great work was done. But more work is expected.

1.In the review for food detection you need to select one class of compounds, for example antioxidants. It has been performed good systematization for phenolic antioxidants and oxidants. But I didn't find in abstract anything about nitrite. This part doen't fit to review. You sould specify the target species in abstract.

2.In introduction to antioxidants I suggest to suppress text, and provide info about type of antioxidants, issued in the paper. I was confused to read a lot about vitamins and then to find that CA and caffeic acid are phenolic antioxidants. Also Figure 2 with sensor should be moved to appropriate part, 

3.Electrochemical methods description must be suppressed. Please, provide useful information, like which parameters are measured and analyzed.

4.References. In review it is better to reduce ratio of references 2008-2014 should be reduced compared to 2017-2020

5.English should be carfully checked.

In conclusion. Paper can be accepted after some corrections

Author Response

I have uploaded a correspondig pdf file

Reviewer 2 Report

The manuscript presents a review in the field of nanomaterials in electrochemical sensing area: applications and challenges in food analysis.

The target analytes included in the abstracts are not well choose: ascorbic acid (AA), phenolic antioxidants, caffeine (CAF), caffeic acid (CA), and nitrite.

CA is a phenolic antioxidant also. Why nitrite was included?

Carbon nanotubes (CNTs) and gold nanoparticles (AuNPs) are not, in my opinion, the most used nanomaterials in the development of electrochemical sensors. If the review is about these materials the title, the abstract and the introduction must be fundamentally modified.

The section Electrochemical techniques is not relevant for this review. Furthermore, the presentation is not at the level of this journal.

3.1 Carbon-based nanomaterials section is not clear. The details about the materials are scarce and incomplete. The inclusion of data about sensors detecting metal ions such as Cd2+, Pb2+, Cu2+ and Hg 2+, for example, do not have any sense.

New reviews in the field from 2020 are not included.

The same situation is for section related to gold nanomaterials.

3.3 Hybrid nanocomposites includes some information about conducting polymers and chitosan. But the sensors included here are inappropriate regarding the theme of the review. Why, for example, based on polypyrrole–chitosan–titanium nanocomposite for the glucose detection is presented? What is the relationship with the theme of the review?

Presentation of Phenolic antioxidants is not relevant. Why other antioxidants are discussed here?

4.2 Caffeic acid must be a subsection of the Phenolic antioxidants.  Novel papers in the field, from 2020, are not

In Table 1 the column 1 must be split in two, sensitive material and detection technique.

The same situation for the Table 2.

4.3 Chlorogenic acid must be subsection. This compound was not included in the list from the abstract.

Why 4.4 Rosmarinic acid is included? Why not ferulic acid or gallic acid?

The detection principle of caffeine and other compounds must be included in the manuscript.

The section Ascorbic acid is incomplete. It not include a Table with sensors used for detection of this analyte. A review can be done only for this analyte.

Nitrite is an outlier here. The detection in waters is not the theme of this review.

The conclusion are not relevant. Future trends and perspectives must be included.

The English must be corrected. A lot of typo errors are found along the manuscript.

Author Response

I have uploaded the corresponding pdf file

Reviewer 3 Report

An interesting review discussing the use of nanomaterials in electrochemical sensors,  requiring just a few minor modifications.

It would be interesting if the structures of some of the analytes, caffeic acid,  chlorgenic acid, rosmarinic acid and the other analytes could be included as a figure.

Minor errors – there are a few grammatical errors,  some of which I list below,  these could be corrected and perhaps a thorough read through to check for others.

Line 93 Warbug should be Warburg

Line 104 measured varying should be measured whilst varying

Line 140 such graphene should be such as graphene

Line 294 sensors for the food analysis should be sensors for food analysis

Line 367 as phenolic antioxidant should be as a phenolic antioxidant

Line 462 SrV2O6 needs subscripts

Line 499 sheetsnanocomposite should be sheets nanocomposite

Line 685  energetic drinks should be energy drinks

Line 911 MOFs was calcined should be MOFs were calcined

Author Response

I have added the corresponding pdf file

Round 2

Reviewer 2 Report

The manuscript was improved after revision. However few minor corrections are still necessary before publication

  • Lines 17-18: „metallic nanoparticles and nanomaterials” must be replaced with „metallic nanomaterials”
  • Line 18: „molecules” must be replaced with „analytes”

Typo errors are still present in the manuscript. Some subscript in the chemical compounds must be corrected ( i.e. line 461).

Author Response

I have uploaded my reply as pdf file
